# Land tenure drives Brazil's deforestation rates across socio-environmental contexts

Andrea Pacheco [1,2] & Carsten Meyer [1,3,4]

Many tropical forestlands are experiencing changes in land-tenure regimes, but how these changes may affect deforestation rates remains ambiguous. Here, we use Brazil's land-tenure and deforestation data and quasi-experimental methods to analyze how six land-tenure regimes (undesignated/untitled, private, strictly-protected and sustainable-use protected areas, indigenous, and quilombola lands) affect deforestation across 49 spatiotemporal scales. We find that undesignated/untitled public regimes with poorly defined tenure rights increase deforestation relative to any alternative regime in most contexts. The privatization of these undesignated/untitled lands often reduces this deforestation, particularly when private regimes are subject to strict environmental regulations such as the Forest Code in Amazonia. However, private regimes decrease deforestation less effectively and less reliably than alternative well-defined regimes, and directly privatizing either conservation regimes or indigenous lands would most likely increase deforestation. This study informs the ongoing political debate around land privatization/protection in tropical landscapes and can be used to envisage policy aligned with sustainable development goals.

Tropical deforestation, mostly via conversions of forestlands to agriculture or other human-dominated systems, causes widespread degradation of biodiversity[1] and carbon stocks[2]. Land-tenure rights regulate how and by whom tropical forestlands can be used, and are thus central to deforestation-related sustainability challenges[3]. Land-tenure rights are also fiercely contested, leading to shifts in land-tenure regimes in many tropical forest nations[4]. On the one hand, governments place public lands under protection or respond to land claims of indigenous groups, local communities, or landless settlers[5,6]. On the other hand, private-tenure rights are promoted by liberalizing state control and opening various land-based sectors to privatization[7] or restricted through land reforms or environmental policies[3].

Here, we define 'land-tenure regime' as the combination of tenure-related governance factors that exist over a given parcel of land and are stable over a certain period of time. This includes the 'bundle of rights' associated with the respective tenure category (Supplementary Table 2), but also the implications that these rights may have for tenure

security, as well as the tenure category's predisposition for being subject to particular types of policies or regulations. The shifts in land-tenure regimes resulting from land-rights interventions may have long-run impacts on deforestation rates. Diverse interest groups use claims of improved forest conservation to promote different—often mutually conflicting—tenure interventions ranging from privatization to recognition of communal rights. Policy-makers deciding on these politically charged processes require robust information on the most likely, long-term effects of different interventions on forests. In particular, government programs and NGOs need transferable knowledge to design robust overall strategies with respect to different land-tenure forms or interventions, especially in many tropical regions where the capacity for context-specific assessments is often limited.

However, scientific insights remain ambiguous. Firstly, theoretical predictions on the effects of different land-tenure regimes often contradict one another (Table 1, Supplementary Table 1). Secondly, partly due to data limitations[8], empirical synthesis has been constrained to

[1]German Centre for Integrative Biodiversity Research (iDiv) Halle-Jena-Leipzig, Puschstrasse 4, 04103 Leipzig, Germany. [2]Institute for Food and Resource Economics, University of Bonn, Bonn, Germany. [3]Institute of Geosciences and Geography, Martin Luther University Halle-Wittenberg, Halle (Saale), Germany. [4]Institute of Biology, Leipzig University, Leipzig, Germany. ✉ e-mail: andrea.pacheco@ilr.uni-bonn.de; carsten.meyer@idiv.de

**Table 1 | Exemplary hypothesized deforestation effects of different tenure regimes and regime changes**

| Tenure regime/ regime changes | Predicted long-term effect | Hypothesized mechanisms |
|---|---|---|
| Leaving public lands undesignated to any use, and untitled (if occupied) | Deforestation-inhibiting | Undesignated/untitled status inhibits forest-displacing land-use activities, both because untitled settlers cannot easily access credit and because the uncertainty regarding applicable regulations discourages outside investments, making these lands de facto reserves[53,54]. |
| | Deforestation-promoting | Undesignated/untitled lands lack both clear supervisions by any designated agency[55] and effective exclusion rights. As a result, they often become de facto open-access environments and, as such, are prone to unsustainable exploitation by rational-strategic agents[56–58]. Governments rarely place restrictions on deforesting undesignated/untitled public lands—or even incentivize it by granting claims based on prior clearance[59], or by allowing settlement conditionally on putting the land to productive use[60]. Due to relatively higher land prices for existing private lands on formal markets, poor small-holders or landless individuals searching for land may see themselves forced to clear undesignated/untitled lands at the development 'frontier'[61]. |
| Replacing undesignated/untitled with private tenure through registration, regularization, or titling | Deforestation-inhibiting | Being granted private-tenure rights incentivizes settlers to make longer-term investments in forest-conserving land uses because the extensive exclusion and due-process rights of private landholders reduce their risk of financial default through outside invasion or government seizure[56], thus providing assurance that they will be the sole beneficiaries of their investments. Private titles enable improved enforcement of environmental policies as they facilitate holding specific individuals accountable for complying with environmental obligations[24], such as the obligation to retain certain amounts of forest under Brazil's Forest Code. |
| | Deforestation-promoting | The lower default risk combined with comprehensive withdrawal and alienation rights of private-tenure regimes sparks investments in forest-displacing activities[62]. For example, private landholders can more easily access credit to expand their agricultural fields by using land as collateral[54]. Similarly, under functioning land markets, sell and lease rights will result in an eventual transfer of land to whoever can use it most profitably, which will most typically be an agricultural use[63]. |
| Recognizing claimed land rights of indigenous or local communities | Deforestation-inhibiting | Communities collectively holding land typically create societal rules about resource use. Community members tend to follow these to avoid social exclusion, leading to reduced degradation of communally regulated forest resources[64]. |
| | Deforestation-promoting | Communities will often fail at effectively managing common forest resources, due to different impediments to collective action, such as free-riding and conflicting interests[65]. |
| Privatizing any lands under statutory public ownership, including those under indigenous or conservation regimes | Deforestation-inhibiting | Public institutions often provide ineffective forest governance, e.g., due to limited monitoring and enforcement capacity, high corruption[55], or liberal granting of use concessions for short-term state revenues[64]. Even those publicly owned forests that are under private or community-based management will not be used sustainably in countries with a history of short-lived government institutions, as government proposals for sustaining these resources for long-term benefits will lack credibility[66]. Privatization of public lands promotes the more sustainable, productive use of natural resources by enabling more agile, innovative, and thus effective use at the production margin[55] and internalizing long-term costs of degradation into decisions[67]. |
| | Deforestation-promoting | Individual tenure regimes fail to fully internalize non-monetary (e.g., biodiversity, cultural) or future values of forest resources that accrue mainly to society, rather than the individual. Thus, state-controlled forest governance is necessary for maintaining forests where this is not the most profitable land-use form[55]. |

For a given tenure regime or regime change, both deforestation-promoting and deforestation-inhibiting effects may be expected via different, often non-mutually exclusive, causal mechanisms. A broader overview of hypotheses, with reference to the bundles of rights associated with tenure regimes that mediate these mechanisms, is provided in Supplementary Tables 1–2.

meta-studies across case studies of limited comparability[9–11], and to large-n but single-scale studies focused on one or few tenure regimes[12–14]. To date, systematic large-n assessments of the effects of alternative tenure regimes on deforestation across different scales or regional and temporal contexts are lacking, hampering robust generalizations on the most likely long-term effects of land-tenure policies.

Here, we provide such systematic testing and synthesis of land-tenure effects on tropical deforestation across different spatio-temporal contexts (see Methods; details in Supplementary Information). We analyzed 33 years of agriculture-driven deforestation across Brazilian forestlands, which harbor the world's largest biodiversity and living carbon stores, but are under pressure from ambitious agroeconomic development[15,16]. We capitalize on Brazil's uniquely comprehensive data on both land-tenure[17] and land-use changes[18] and use quasi-experimental approaches to quantify deforestation effects (Methods). To explore likely long-term deforestation effects of land-tenure shifts in tropical regions resulting from major intervention trends such as (re)designation of public lands, communal or private titling, registration, or privatization, we compare six alternative tenure regimes against two counterfactuals, (i) undesignated and untitled public lands with poorly defined tenure rights (hereafter 'undesignated/untitled'), and (ii) individually held private lands (hereafter 'private').

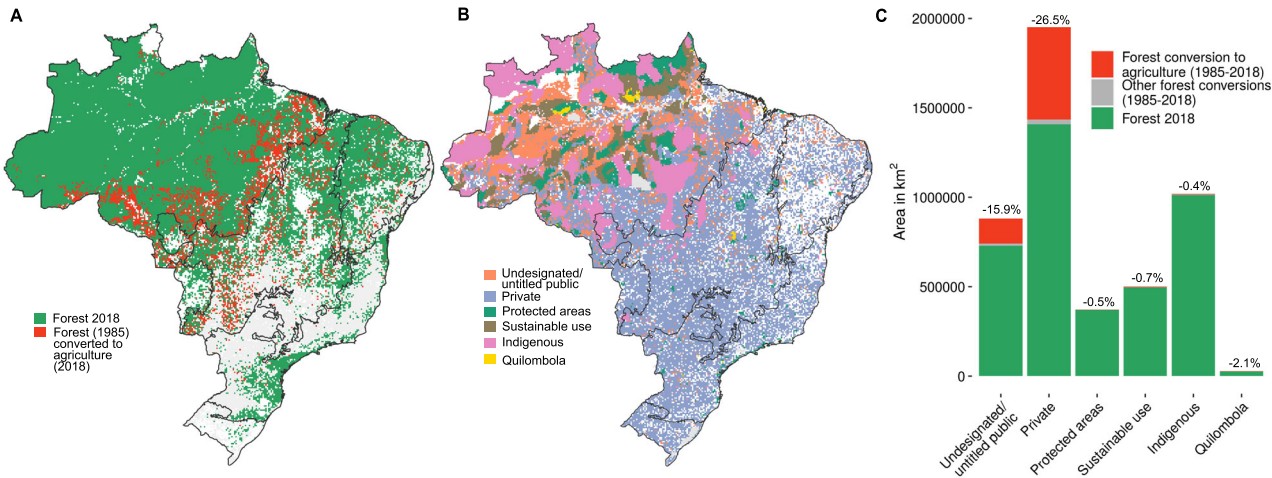

**Fig. 1 | Forest conversion to agriculture (1985–2018) and spatial distribution of different land-tenure regimes in Brazil. A** Shows all forest cover (including natural forests, plantations, savannas, and mangrove tree cover) converted to farming (pasture, agriculture, annual perennial, and semi-perennial crops, including mosaics of agriculture and pasture)[18]. **B** Shows the spatial distribution of six different land-tenure regimes, collated from Imaflora's Atlas of Brazilian Agriculture[17]. **C** Shows total areas of forest that were converted to agriculture (red) or other land uses (gray) between 1985 and 2018, and remaining forest cover in 2018 (green), across all Brazil-wide parcels under each tenure regime. Percentages of total original (1985) forest cover per tenure regime that were converted to agriculture by 2018 are indicated above each bar.

## Results and discussion

### Poorly defined tenure drives deforestation across spatio-temporal scales

We found that 17.4% of Brazil's originally forested 30-m pixels lost forest to agriculture between 1985 and 2018 (Fig. 1A). The vast majority of this deforestation occurred on private (78%) and undesignated/untitled lands (19%; Fig. 1C). The latter are publicly owned lands with poorly defined tenure rights that are not yet designated to any use but may be inhabited by rural settlers without a formally recognized land claim or title. Such undesignated/untitled tenure regimes cover vast areas across the tropics, and in Brazil alone, account for almost one hundred million hectares (963,357 km²; ref. 19), an area larger than Tanzania (Fig. 1B). Different hypothesized mechanisms may drive deforestation under such undesignated/untitled tenure regimes up or down (Table 1, Supplementary Table 1). Here, we aimed to test the predominant prediction that such regimes cause increased agriculture-driven deforestation.

To this end, we used a quasi-experimental study design that combines matching with a generalization procedure to estimate the average treatment effects (ATE) of undesignated/untitled regimes on deforestation in Brazil. For this, we first matched land parcels under alternative tenure regimes to undesignated/untitled land parcels. We used matching covariates known to influence deforestation to broadly capture factors that are likely to be relevant for policymakers when deciding on shifts in tenure regimes. After dropping the unmatched land parcels, we assessed in how far effects estimated with the remaining, matched-data subsets would be generalizable to the entire population of land parcels, using Tipton's index of generalizability (T-index), a metric that captures similarities across covariates in different populations. In order to broaden the generalizability of our results, we generated weights for each parcel in each matched subset to more closely represent the covariate distribution of the entire "population" of land parcels. Subsequently, we estimated population-wide effects via regression analyses while explicitly incorporating generated weights (details in Methods and Supplementary Information, full results in Source Data file).

Our Brazil-wide analyses revealed that, on average, undesignated/untitled regimes increased deforestation between 1985 and 2018 by ~12.4–23.2% relative to all other tenure regimes (Fig. 2A, large circles).

To assess the consistency and potential transferability of these results across different contexts in the tropics, we repeated these quasi-experimental tests for 48 different combinations of narrower spatial and/or temporal extents. These extents correspond to highly distinct socio-environmental contexts, characterized by different bioclimatic regions with distinct agricultural sectors and environmental governance regimes, as well as by different historical time-periods since the mid-1980s defined by major macro-economic events, national policies, and deforestation highs or lows (details in Supplementary Information). These tests revealed higher deforestation under undesignated/untitled compared to the respective other tenure regimes in 141 out of 196 cases (lower deforestation in six cases, nonsignificant in 49, Supplementary Fig. 1, Source Data file, and Supplementary Table 6). These results were qualitatively robust to weighting all cases by balance levels of their respective datasets post-matching, and to filtering out strictly protected and sustainable-use protected areas that were only officially established after the beginning of the respective time period or had unknown establishment dates. Tipton's generalizability index also indicated that covariate distributions were similar across all alternative tenure-regime comparisons and undesignated/untitled land parcels, meaning these results are highly generalizable to the entire population of land parcels at the respective spatial-temporal scales (see Supplementary Information; Supplementary Figs. 3–4, Source Data file, Supplementary Tables 4–5). Overall, these results provide strong evidence that across vastly different contexts, the lack of well-defined tenure rights on public lands causes increased agriculture-driven deforestation. As such, they substantiate appeals for policy interventions to install alternative tenure regimes[20,21], which may be particularly urgent given the increasing deforestation rates in these lands observed since 2017[22].

### Private tenure decreases deforestation vis-à-vis poorly defined tenure, but less so than the alternative, well-defined regimes

Over recent decades, global development policies strongly promoted placing undesignated/untitled public lands under private-tenure regimes[23] through tenure interventions such as regularization, titling, or registration. Conservation and sustainable-development organizations alike commonly support such interventions[24], hoping that associated improvements in tenure security and clarity will promote more

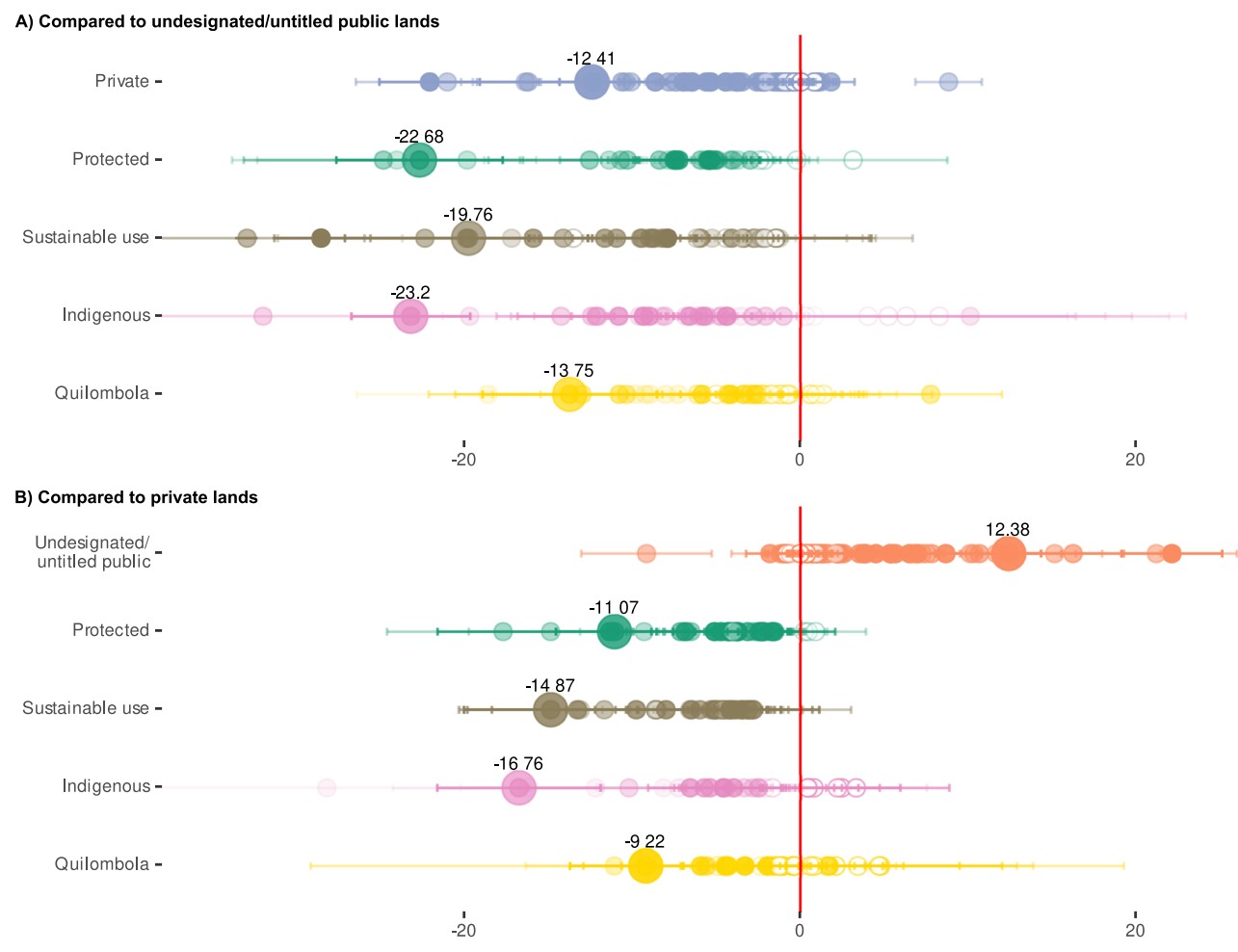

**Fig. 2 | Effects of alternative land-tenure regimes on forest-to-agriculture conversion rates in Brazil.** Circles indicate average effects sizes estimated using regression analysis (using matched parcels) at different spatial-temporal scales, compared to two alternative counterfactuals: **A** undesignated/untitled public lands with poorly defined tenure rights, and **B** private lands. Labelled effect sizes (larger circles) report effects across Brazil over the time period 1985–2018. Effects to the left of the zero line indicate a decrease in average parcel-level deforestation rate (to the right: increase). Filled circles indicate statistically significant effects ($p < 0.05$; nonfilled: not significant); upper/lower confidence intervals are plotted to the left/right of each circle centroid. Higher transparency of filled circles indicates high levels of imbalance in the matched dataset (multivariate imbalance measure $L_1$). See Supplementary Figs. 1–2 for detailed presentation of scale-specific results for all tenure regimes. Source data are provided as a Source Data file, where results from time-filtered robustness tests are also found.

sustainable resource management—although shifts to private regimes may also promote deforestation via other mechanisms (Table 1, Supplementary Table 1). The relative importance of these deforestation-promoting and -inhibiting mechanisms is likely context-specific. To guide more general policies, an important first step is thus to quantify their combined net effects and how consistent these effects are across different contexts.

Similarly to how we analyzed the effects of undesignated/untitled tenure, we thus assessed the directionality, magnitude, and consistency of net effects of replacing undesignated/untitled tenure with private tenure across the 49 distinct spatiotemporal scales. In our quasi-experimental analysis setup, private tenure would have caused a 12.4% average reduction in the deforested area compared to the matched parcels under undesignated/untitled tenure across Brazil over the period 1985–2018 (Fig. 2A; note that these analyses are not confounded by differing initial forest covers; see Supplementary Information, Fig. 5). Yet, these deforestation-reducing effects were not consistent across narrower regional-historical contexts. At these narrower scales, net effects of private tenure were deforestation-decreasing in only 61.7% of cases (63.2% if balance-weighted, deforestation-increasing: 8.5%/8.2% if weighted, nonsignificant: 29.8%/

28.6%; Supplementary Figs. 2, 4, Supplementary Table 6). These findings indicate that the environmental benefits of tenure interventions promoting private rights over undesignated/untitled lands more often outweigh the risks than vice versa. Yet, they also suggest that private tenure does not reliably lead to improved forest outcomes. Indeed, recent titling activities in Brazil's Amazon region have caused deforestation increases in the years immediately following the interventions[13], highlighting the importance of coupling titling interventions with environmental policies to effectively safeguard forests.

Beyond private tenure, different interest groups advocate for various other regimes with different but similarly well-defined tenure rights to replace undesignated/untitled regimes, including indigenous, community-based, strict-protection, and sustainable-use protection regimes (Supplementary Table 2). We assessed which of the alternative regimes could reduce deforestation most effectively and reliably. To this end, we compared the effects of these alternative tenure regimes against an undesignated/untitled counterfactual across 34 different scales (Supplementary Information). To enable indirect comparisons of the performance of the alternative regimes, we weighed the undesignated/untitled counterfactuals to represent the covariate distribution in the entire population of parcels at each respective scale, which

effectively standardized the counterfactuals across the tenure-regime comparisons (Supplementary Methods 3.4). These tests revealed that under most regional-historical contexts, private tenure under-performed all alternative regimes in protecting forests with the exception of quilombola regimes (privately owned lands of communities of self-identified descendants of Afro-Brazilian slaves; see also next section). Specifically, private tenure had the highest risk among all alternative regimes of increasing deforestation over the undesignated/untitled counterfactual (8.8% of scales considered; 8.4% if balance-weighted), and was least likely to cause high deforestation reductions (2.9%; 2.2% if balance-weighted), and was second-most likely to cause the lowest reductions/highest increases (after quilombola, 26.2%; 28% if balance-weighted; Supplementary Table 4). Overall, these results suggest that among the alternative tenure interventions that might reduce the deforestation associated with undesignated/untitled tenure by installing better-defined tenure rights, interventions leading to private tenure would be the least reliable and typically among the least effective options across vastly different socio-environmental settings.

### Protection-oriented tenure regimes reliably decrease deforestation, while the effects of indigenous and community-based regimes are ambiguous

We expected that strict-protection and sustainable-use protection regimes would reduce deforestation most strongly, as the associated bundles of rights are specifically designed for conservation purposes (Supplementary Tables 1–2). Fully protected areas, in particular, remain the mainstay of global conservation strategies, despite concerns about inconsistencies in management effectiveness[25,26] and debate about the extent to which the conserved natural resources should be open to sustainable use[27,28]. Our results support our hypothesis, in that strict-protection and sustainable-use regimes had, respectively, the second- and third-strongest deforestation-reducing effects at large scales (Fig. 2/Supplementary Fig. 1). The two regimes also most consistently achieved at least some reduction in deforestation across the narrower regional-historical contexts (88.2% and 76.5% of cases with significant negative effects, respectively, Supplementary Table 4). The above results were robust both to weighting by balance post-matching and to filtering later-established conservation areas (Supplementary Fig. 3A, Source Data file, Supplementary Tables 3–4, see Supplementary Methods). However, sustainable-use regimes were about five times more likely to outperform than to underperform alternative regimes in protecting forests (largest/smallest deforestation reductions in 41.2/8.8% of cases; 42.6/7.4% if balance-weighted; 47.4/9.2% if time-filtered), this relative performance was much less clear for strict-protection regimes (26.5/14%; 26.3/10.6%; 15.7/7.9%; Supplementary Tables 4–5; note these differences were not confounded by protected-area siting[29], see Supplementary Methods, indirect comparisons of relative effects are based on standardized counterfactuals, and T-index scores were all ≥0.5, indicating effect estimates are generalizable to the entire population, Source Data file). This aligns with findings from other tropical countries that stricter protection does not necessarily increase conservation outcomes[30]. Moreover, this indicates that while any conservation-focused regime may reduce deforestation more reliably than alternative regimes under very different contexts, specifically sustainable-use protection regimes may most reliably achieve large reductions across contexts.

We also analyzed the effects of tenure held by indigenous peoples and local communities (IPLCs). IPLCs have recognized tenure rights over a large and growing portion of the world's forestlands[5], and are increasingly embraced by environmental policies as critical partners for conserving biodiversity and carbon[31]. Provided that IPLC land claims exist, IPLC-tenure rights might be recognized over any land. Thus, we assessed effects against both undesignated/untitled and private-tenure counterfactuals. Our results showed that against either

counterfactual, both indigenous and quilombola tenure regimes decreased Brazil-wide deforestation during 1985–2018 (Fig. 2/Supplementary Fig. 1). Yet, our tenure-regime comparisons across the different spatial-temporal scales yielded inconsistent results. Significant deforestation-reducing effects only emerged in 58.3–59.8% of these comparisons (depending on balance-weighting; Supplementary Table 4). The only specific comparisons that fairly consistently showed deforestation-reducing effects of IPLC tenure were those of indigenous tenure vis-à-vis an undesignated/untitled counterfactual (76.5–82.8% of cases, Table S6). Indigenous tenure reduced deforestation vis-à-vis private tenure in only 59.4–70.4% of cases, although true population-wide effects might be even more ambiguous. This is because these latter effects were only generalizable from matched parcels to larger parcel populations in 17% of cases, mostly in the Cerrado biome (T-index ≥ 0.5 in Source Data file), reflecting the biased siting of indigenous reserves in other biomes, specifically in areas farther from cities and at higher elevations, relative to the population averages (Supplementary Table 8). Quilombola tenure, in turn, reduced deforestation least reliably and often least effectively among the compared tenure regimes against either counterfactual, notably lacking significant effects during most periods in Caatinga—the biome where most quilombola lands are situated (Supplementary Figs. 1, 4). These ambiguous results on the effects of community-based tenure regimes on deforestation rates are in line with diverging theoretical arguments (Table 1; Supplementary Table 1). Overall, the limited generalizability and transferability of IPLC-tenure effects on deforestation rates evident in our results suggest that synergies between IPLC tenure and forest conservation objectives may indeed arise in diverse contexts. However, designing policies with these synergies in mind will likely require detailed contextual knowledge to ensure IPLC-tenure interventions have positive forest outcomes.

### Benefits of private ownership vs. public reserve regimes for protecting forests, and the special context of Amazonia

While we designed our analysis and cross-contextual synthesis approach to identify consistent (and thus potentially transferable) effects across diverse social-environmental settings, we found important divergences from overall effects for Amazonia, where 90.5% of Brazil's remaining undesignated/untitled forest is situated (Fig. 1). Here, all three public reserve regimes (strict protection, sustainable-use, and indigenous) had consistently weaker deforestation-reducing effects vis-à-vis undesignated/untitled regimes than quilombola tenure (communal yet private regimes; Supplementary Fig. 1). Even more surprisingly, private tenure changed from being deforestation-increasing vis-à-vis an undesignated/untitled regime in 1985–1990 to being the second-most (after quilombola) or most strongly deforestation-decreasing regime from the early 2000s (Supplementary Fig. 1). Both results were robust to balance-weighting, not confounded by systematic differences in initial forest cover (Supplementary Fig. 7), and were highly generalizable to the entire Amazonian population of undesignated/untitled and private land parcels (Tipton's index ≥ 0.8; Source Data file).

These counter-intuitive Amazonian effects might be explained by the region's specific environmental governance setting. Over recent decades, Amazonian private landholders have been subject to stricter forest-protection policies than those in other biomes, including the Forest Code's requirement to retain four times the forest cover required in other biomes, as well as earlier-implemented soy and beef moratoria[32,33]. At the same time, understaffing and logistic difficulties due to Amazonia's remoteness may disproportionately hamper the effectiveness of government policing of the region's public reserves[34]. This lower capacity for law enforcement in public areas, in turn, may reduce the difference in the de facto governance of Amazonia's undesignated/untitled lands vs. its conservation/indigenous lands (relative to differences in other biomes). This could mean that for

remote public lands with poorly defined tenure rights and limited public capacity for on-the-ground control, privatization that is strongly coupled with extensive environmental obligations[35] might be effective in reducing deforestation, as it partially transfers responsibility and accountability for forest governance from public institutions to specific individuals. Moreover, this suggests that there is potential to substantially decrease future Brazil-wide deforestation rates by extending more stringent private-actor-focused environmental policies (e.g., as currently via the Forest Code for Amazonia), to Brazil's other remote biomes, where remaining forestlands are mostly private (Cerrado: 80.4%; Pantanal: 92.8%; Fig. 1B, C).

These findings for Amazonia also raise the broader question of how more general privatization of any publicly owned lands in the tropics might affect deforestation rates. Globally, over 70 percent of forestlands, including most indigenous and conservation lands (as well as undesignated/untitled lands), are statutorily owned and administered by public institutions[36]. Different hypotheses predict that replacing public with private tenure would reduce deforestation, fueling arguments for liberalizing state control over these lands (notwithstanding counter-hypotheses; Table 1; Supplementary Table 1). Our systematic tests comparing matched parcels under alternative public regimes against private parcels did not find support for a general public-private dichotomy (Fig. 2B). Instead, they showed that replacing any public regime other than undesignated/untitled with private tenure would have likely increased deforestation in most regional-historical contexts, even when solely counting generalizable cases (i.e., 66.7% of country-wide, 77.8% of biome-specific long-term, and 75% of biome-specific short-term tests; mean effects ranging from 1.6 to 28.2% deforestation increase; results qualitatively robust to balance-weighting and time-filtering; Fig. 2B, Supplementary Figs. 1, 3–4; Source Data file). In fact, despite our earlier findings that private tenure more effectively reduced recent deforestation on Amazonian undesignated/untitled lands than public reserve regimes, directly replacing those alternative public regimes with private tenure would have most likely increased deforestation in Amazonia, particularly after the year 2000 (60.7% of all tested time-periods, 80% after 2000; Source Data file). This apparent paradox indicates that privatization may only effectively counter the specific deforestation mechanisms acting on Amazonian undesignated/untitled public lands—but not those on state-protected or indigenous lands. Informing current political debates, these insights indicate that the privatization of protected or indigenous lands would likely increase deforestation in Amazonia, or elsewhere[16,37].

In summary, against a backdrop of oftentimes ambiguous empirical evidence, theories, and interest groups' claims, our study can shed new light on the direction and relative magnitude of the net effects of alternative land-tenure regimes on tropical deforestation. We achieved this through systematic quasi-experimental testing, using weights to generalize our results to the entire population, and synthesizing results across different spatiotemporal scales and contexts. Our results may inform environmental practitioners about the likely environmental impacts of different land-tenure regimes. Moreover, they may offer guidance to policymakers about which alternative tenure interventions might reduce long-term deforestation rates most effectively and reliably under different socio-environmental settings. This can help clarify how different tenure policies might align or misalign with forest-dependent sustainable-development goals such as climate change mitigation and biodiversity conservation.

Despite the context-specificity of human-environment systems[38], we could derive several conclusions that were consistent across highly diverse environmental, socio-political, and economic contexts in Brazil. These highly consistent results may be more likely than others to also hold for yet other tropical contexts, and, therefore, may be most relevant to other countries that model their forest-governance policies after those in Brazil[39,40]. In particular, placing undesignated/untitled public lands with poorly defined tenure rights under any other tenure regime will likely substantially reduce deforestation. Reducing deforestation appears most probable when implementing conservation-focused regimes, where sustainable-use regimes, in turn, appear more likely to cause large reductions. Large reductions are least certain when promoting private land rights, although our more context-specific Amazonian results indicate that this can be highly effective where there are constraints to on-the-ground government control and if private rights are coupled with extensive environmental obligations. Finally, privatizing public lands other than undesignated or untitled, such as protected areas or indigenous reserves, will most likely increase deforestation. For those tenure regimes for which our assessment does not indicate high generalizability or consistency of effects across scales, such as IPLC-based regimes, guidance to sustainability policies should be based on further research into the context-distinguishing factors. Expanding the systematic cross-scale testing shown here to other tropical regions will be contingent on governments making parcel-level land-tenure information more accessible. Greater transparency is particularly crucial with regard to private and IPLC-tenure rights, which cover much of the remaining tropical forest estate but showed the most context-dependent effects.

## Methods

### Tenure data

We used the comprehensive, publicly available data on land-tenure categories compiled by Imaflora (v.1812[17]) for 83.4% of the Brazilian territory, which is based on 18 official sources, and was integrated using an expert-vetted system to systematically resolve data conflicts resulting from, e.g., overlapping land claims due to illegally fabricated land titles or mapping errors[19] (Supplementary Methods, sections 1 and 2.1). For most tenure categories, the available data lack, or have incomplete information on the date of each parcel's formalization (i.e., titling or demarcation). Despite possible changes in official ownership status, it can be assumed that for the majority of parcels, the basic type of tenancy (e.g., public institutions vs. indigenous communities vs. private individuals) did not change over the course of our study period. However, as this assumption could be problematic for certain tenure categories, we took several steps to minimize possible bias in our statistical analyses and conclusions.

Firstly, we performed all analyses over multiple spatial and temporal extents and assessed whether results for Brazilian subregions and time-periods with known changes in tenure patterns were qualitatively consistent with those for 'tenure-stable' regions/periods. Secondly, we excluded tenure subcategories defined via programs that only came into existence after our study periods began (e.g., all Terra legal parcels were excluded from the analyses). Thirdly, we performed robustness tests for selected tenure categories with documented 'treatment' dates, where we filtered out parcels for which today's tenure category was non-existent or unclear at the beginning of the respective study period. Fourthly, we assessed possible biases in our quasi-experimental setup due to remaining statistical imbalance, omitted variables, and systematic differences in initial forest cover between 'treatment' and 'control' units. Specific steps are further outlined in our description of the tenure categories analyzed below (also see Supplementary Methods section 2.2) and of our study design and statistical approach (also see Supplementary Methods sections 3.3–3.6).

We grouped several Brazil-specific categories to correspond to general tenure categories present in most tropical forest nations. Private tenure ('private') was defined as properties with individual ownership, and we included properties from different sources (CAR, SIGEF) but excluded all properties titled by the Terra Legal program, as it only began operating in 2009. Note that deforestation effects of property titling under the Terra Legal program were recently the focus of different study[41]. Undesignated and untitled public lands

('undesignated/untitled') were defined as those publicly owned, yet not formally assigned to any purpose or with otherwise poorly defined tenure rights. We merged public properties listed in the Imaflora dataset as either 'undesignated lands' or 'rural settlements' into this category but excluded all rural-settlement parcels that are part of the Terra Legal program, as the program may bias deforestation behavior in anticipation of a land title[41]. We followed the categorization of Brazil's Ministry of Environment for conservation-focused tenure regimes, distinguishing strict-protection ('protected areas') from sustainable-use protected regimes ('sustainable use'). Areas of environmental protection ('Áreas de Proteção Ambiental') are excluded from the Imaflora dataset, and thus not included in this analysis[17]. We maintained three categories for indigenous or local community-based (IPLC) tenure regimes, ('indigenous', 'quilombola', and 'communal') given the differences in their histories, legal statuses, and their bundle of rights (Supplementary Methods, section 2.2). Communal lands were excluded from the main results reported here due to the heterogeneity of their bundle of rights, and because there were insufficient recorded communal land parcels to support our analyses in all biomes except for Amazonia. Results for communal regimes are provided in Supplementary Figs. 1–2, and Source Data file.

## Forest cover and covariate data

We used the 30-m-resolution annual land-cover/use data provided by Mapbiomas[18] for our calculations of forest-to-agriculture conversion rates (Supplementary Methods, section 2.3). We used a set of covariates known to influence forest-to-agriculture conversion that is likely to be relevant for policymakers when deciding on shifts in tenure regimes. These include market accessibility (represented by travel time to the nearest city[42]) and agricultural suitability (represented by slope and elevation[43];). Both of these variables strongly determine achievable land rents and thus the opportunity costs of 'assigning' parcels to particular tenure regimes, while also capturing the inherent bias of the siting of different tenure regimes[29]. We also included human-population density[44] as larger populations can more strongly influence policy processes for the formalization of property rights (e.g., via titling of private regimes or recognition of IPLC land claims), whereas lower population density implies more liberty to create conservation regimes or leave land undesignated. Finally, we included parcel area in ha[17], because property size influences the prices landholders pay for receiving land titles[41], as well as specific forest/agricultural policy requirements and levels of compliance to these policies[45] (see details in Supplementary Methods, section 2.4).

## Study design

Our goal was to assess and synthesize the direction, strength, and consistency of the longer-term effects that plausible shifts between alternative land-tenure regimes would have on agriculture-related deforestation rates in Brazil. Rather than quantifying near-term impacts of specific tenure-intervention events such as titling, we thus wanted to capture the differential forest-to-agriculture conversion rates under alternative land-tenure regimes over periods of several years to decades (Supplementary Methods, section 3.1). Moreover, we wanted to evaluate the extent to which the deforestation effects of these tenure-regime differences might apply across diverse socio-environmental settings within Brazil, and, thus, potentially transfer to other tropical forest regions. To this end, we systematically tested effects across 49 different combinations of spatial and temporal extents that correspond to highly diverse regional and historical environmental, socioeconomic, and policy contexts (i.e., across Brazil's entire territory and its biomes Amazônia, Caatinga, Cerrado, Mata Atlântica, Pampa, and Pantanal, and across our entire study period 1985–2018 and subperiods 1985–1990, 1991–1995, 1996–1999, 2000–2004, 2005–2012, and 2013–2018; Supplementary Methods, section 3.2).

## Matching

For each of these scales and tenure-regime comparisons, we tested effects using a quasi-experimental study design (Supplementary Methods, section 3). We first applied coarsened-exact matching implemented in the 'cem' package[46] in R (versions 3.5.1–4.0.2)[47], which involves temporarily 'coarsening' each confounding variable into bins (predetermined strata), and dropping unmatched observations from the sample. We used automated coarsening for elevation, slope, and human-population change, but manually defined bins for travel time to the nearest city and for parcel area. We divided travel time to nearest city into bins of 0–2, >2–6, >6–12, >12–24, and >24 h, and parcel area into 14 bins of 0–2, >2–5, >5–15, >15–50, >50–100, >100–500, >500–1000, >5000–10,000, >10,000–50,000, >50,000–100,000, >100,000–500,000, >500,000–1,000,000 ha. By conducting CEM individually for each of our defined spatiotemporal extents, we assured exact matching considering the total spatial and temporal variation in the covariates at the respective scale. We use the $L_1$ measure developed by King et al.[46] to calculate the remaining imbalance post-matching. To make cases of high remaining imbalance post-matching easily recognizable, we visualize imbalance as transparency gradients in all plots of estimated effects (Fig. 2, Supplementary Figs. 1–4). Moreover, we explicitly incorporate imbalance into our cross-scale synthesis of results (see Supplementary Methods section 3).

## Estimation of population-wide effects

Post-matching, we faced the limitation that although exact matching using CEM improved the balance in the data and the robustness of estimates, dropping non-matched observations limited the generalizability of effects exclusively to the matched subsample of data. Given our overarching aim to determine the generality of effects, we applied recently developed statistical methods that extend the generalizability of effects from a sample of data to a broader population[48]. Specifically, we first conducted a generalizability assessment of each of these tenure-regime comparisons at each scale considered using the 'generalize' package in R[48]. We calculated Tipton's index of generalizability (T-index), a metric that describes levels of covariate similarity between two groups (i.e., here, between the matched subset of land parcels and the entire population of parcels at a given spatial-temporal scale) (Source Data file). To distinguish cases where matched-data subsets were sufficiently different than the entire population of land parcels, we also calculated the absolute standardized mean difference (ASMD), of each covariate (Supplementary Table 8). These ASMD indicated that matched-data subsets were, on average, at lower elevations, farther away from cities, and in larger areas than the average land parcel in Brazil. We found no systematic differences in these patterns across spatial or temporal scales (Supplementary Table 8).

We generated weights in order for the matched-data subsample to more closely represent the entire population of land parcels. We calculated parcels' weights as the inverse odds of their probability of being matched, meaning that observations with a greater probability of being in the entire population had greater weights. Weights were calculated using Lasso and were incorporated into the estimation of effects.

We estimated effects by fitting generalized linear models (GLMs) with a binomial error distribution and a logit link to the respective matched dataset. We used the uncoarsened variables as model covariates, the previously generated weights to resemble the entire population of parcels, and additionally included the federal state as a fixed effect to control for state-level differences in governance regimes and effectiveness. To control for possibly remaining spatial autocorrelation in model residuals, we clustered our standard errors by municipality (Supplementary Methods, section 3.4). We estimated:

$$\text{logit}(p) = \beta_0 + \beta_1 tf + \beta_2 l + \beta_3 s + \beta_4 tt + \beta_5 pd + \beta_6 r + \beta_7 w + \beta_7 st \quad (1)$$

where $p$ is the per-pixel probability of forest conversion, $tf$ is the tenure regime, $l$ is the average elevation in meters, $s$ is the average slope in degrees, $tt$ is the average travel time to the nearest city in minutes, $pd$ is the average population density, $r$ is the area of the parcel in ha, $w$ are the generated weights, and $st$ the federal state. Note that binomial models of percentage forest loss automatically capture differences in initial forest area, by evaluating the total forest areas (counts of pixels) that were converted to agriculture vs. those that remained. We calculated average marginal effects (AME) using the 'margins' package in R[49], transforming coefficient estimates to average per-forest-pixel probabilities of conversion to agriculture with respect to the tenure form in question[50] (Source Data file). We provide estimates of these effects using both the generated weights as described above, as well as without the generated weights in the Source Data file (see also Supplementary Table 8 for an overview of differences in covariates between matched samples and the entire population of land parcels).

Finally, we tested the sensitivity of our results to potential omitted-variable bias by calculating Rosenbaum bounds (Supplementary Methods, sections 3.4, Source Data file, Supplementary Table 7). We extensively tested the robustness of our results to violations of our constant-treatment assumption and to possible biases due to remaining imbalance post-matching, the differing initial forest cover of treatment and control parcels, and geographical siting of tenure regimes (Supplementary Methods, sections 3.5 and 3.6).

### Consistency of findings across scales

We formally synthesized the estimated scale-specific effects via two complementary approaches. First, we assessed the consistency of the direction of the effects by calculating percentages of scale-specific models with, respectively, significant deforestation-increasing, significant deforestation-decreasing, and no significant effects (Supplementary Tables S4–6). Second, we assessed how consistent the relative rankings of alternative tenure regimes were in terms of the magnitudes of their effects vis-a-vis a given counterfactual, by calculating percentages of scales at which each tenure regime showed higher/lower effects than all others (Supplementary Tables S4–6). Note that, although relative ranks were inherently indirect comparisons of alternative tenure regimes to differently-matched counterfactuals, both the undesignated/untitled counterfactual and the private counterfactuals were weighted to represent the covariate distribution in the entire population of parcels at each respective scale. This weighting thus effectively provided a standardized counterfactual for effect estimations across all tenure-regime comparisons at a given scale.

### Reporting summary

Further information on research design is available in the Nature Research Reporting Summary linked to this article.

## Data availability

All data used in the figures and empirical analyses of this study are publicly available[17,18,42–44]. Processed data from these sources are available as Supplementary Data 1, and full regression outputs and Rosenbaum bounds are available as Supplementary Data 2–3. All data are accessible at (https://doi.org/10.5281/zenodo.7068678)[51]. The data generated in this study on the estimation of effects per each spatio-temporal scale are provided in the Source Data file. Source data are provided with this paper.

## Code availability

All code used for the empirical analyses is available on GitHub (https://github.com/pacheco-andrea/tenure-defor-br)[52].

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

## Acknowledgements

We are grateful to Jan Börner, Andrea Perino, and Martin Quaas for helpful comments on an earlier version of this manuscript, to Marina Jiménez-Muñoz and Jorge Sellare for advice on statistical analyses and methodology, and to Ruben Remelgado for support in data preparation.

A.P. and C.M. acknowledge funding by the Volkswagen through a Freigeist Fellowship to C.M. (A118199), and additional support by iDiv (DFG, FZT-118).

## Author contributions

A.P.: conceptualization, methodology, software, validation, formal analysis, investigation, resources, data curation, writing—original draft preparation, writing—review and editing, visualization, project administration. C.M.: conceptualization, methodology, validation, writing—original draft preparation, writing—review and editing, supervision, project administration, funding acquisition.

## Funding

## Competing interests

The authors declare no competing interests.
