## [Peer Review File · Nature Communications]

Reviewers' Comments:

Reviewer #1:

Remarks to the Author:

Comments

This manuscript takes up a very important question. It also provides quite a lot of solid analyses. I would say, certainly, that this work (and prior works) support the importance of tenure regimes. Also, I definitely feel highlighting undesignated+untitled ('UnUn') lands is a useful policy guide. Consequently, I feel very confident that, in its final version, this effort will contribute quite a lot. That said, I'd also suggest multiple significant additions are required to arrive at its final version. Additions are needed not due to any mistake but rather just challenges inherent in a big question; thus, with only positive intentions I offer some ideas below on refinements and communications.

Contextualization Most confusing for me, despite all the well-motivated and significant effort to distinguish then consider separately quite a number of settings, was perhaps under-publicizing a 'flip' in results for Amazonia (which seemingly is where a lot of these UnUn lands are located?). While I'd agree highlighting the lack of governance structures for UnUn is a really helpful thing, leaning towards single conclusions without conditioning upon contexts that differ enormously in terms of the reach / capacity of public governance – for any given tenure – was confusing for me. (Specifically, e.g., is a protected area in Sao Paulo usually very similar to PAs within Rondonia?)

Having gotten through the paper, I noticed more in the Abstract the "except in remote ..." clause which qualifies the prior statements. Yet in way, I guess I might summarize that "poorly defined" to some extent effectively describes what one was supposed to take away from the Abstract with respect to "well-defined" non-private regimes. Put another way, I think the paper is broadly right to want to include multiple elements in "regime", yet maybe political-economic factors add too? Along these lines, on the positive side I am glad to see Table 1 include stories in either direction. Allowing for more than one empirical conclusion also could be okay, if each fits a given context (by which I mean not each of 50+ combinations but, say, "the governed versus the far frontier"?).

Standardization It is an important point to highlight for readers that matching drops some areas; quite intentionally, in fact, as one endeavors to find control observations in the same contexts as those in which the treatments are found. Thus, e.g., the ATT and ATU impacts could vary, as if treatments are in particular spaces, while impacts differ across spaces, ATT does not generalize. Nonetheless, despite the utility from awareness and communication of these facts, I feel it would be much more direct and clear to simply provide the "In The Context We Found Them" estimate of the impacts of these various regimes. It is for those places that we can really estimate impacts.

Naturally, there are good reasons for standardization, especially for the authors' goals of ranking across regimes/interventions in terms of impact. So the weighted results could also be of interest. Yet I do not feel the average readers will really have any idea about what they are looking at here. Since I think the weighting has some fine justifications and is being done in okay and transparent manner, for those who bother to look, I use the term "addition" for adding unweighted estimates. What might add further, then, is highlighting when and why weighted conclusions have differed? Being more specific, really I'd lead with the Where They Were Found estimates of these impacts. Further, I'd emphasize that non-random sitings probably tell us something about the net benefits. Then if the authors think rankings for outside those sites are useful, explain when rankings shift?

Connection For putting the manuscript's and the above additional points in perspective, I suspect perhaps some additional references to, and connections, to prior literatures could add for readers:

- I'm not in tenure literatures, per se, but my impression is many have been, for some time; for instance, I'll mention William Sunderlin, who seems uncited while it took me no time to find relevant works [listed below & highly cited, even if maybe not his best fits here?]
- I am in impact evaluation, which has been growing for deforestation – but perhaps less in ways relevant for the conservation-policy comparisons here? Yet grabbing just one author cited here,

again it took little time to find comparison of regimes (surely there are more!?)

The reason for connections with literature on tenure and conservation regimes is not only that the past work offers context. Also, a relatively simple extension now becoming more common might contribute. I think that the matching done here is perfectly reasonable for what it is, i.e., some use of fixed characteristics (elevation, slope, travel time), at least for the most part. That surely helps, very much in the spirit of, say, the early Lucas Joppa work on locational bias at even larger scale. However, I think we know now that such matching certainly does not guarantee "parallel trends", while simply extending such matching to pre-treatment deforestation outcomes can help with it. For comparing such different contexts, I'd suspect it might be very helpful to see if this matters.

Two Quick Literatures Examples (Sunderlin for tenure regimes and their consequences, Pfaff for conservation regimes' impacts)

SUNDERLIN QUICK LOOK HERE → MAYBE COULD ADD SOME OTHER LONGSTANDING TENURE FOLKS TOO?

From exclusion to ownership? Challenges and opportunities in advancing forest tenure reform. William D Sunderlin, Jeffrey Hatcher, Megan Liddle 2008 Rights and Resources Initiative

Land tenure and REDD+: The good, the bad and the ugly. AM Larson, M Brockhaus, WD Sunderlin, A Duchelle... - Global environmental change, 2013

Linking forest tenure reform, environmental compliance, and incentives: lessons from REDD+ initiatives in the Brazilian Amazon. AE Duchelle, M Cromberg, MF Gebara, R Guerra... - World Development, 2014

Incorporating land tenure security into conservation. Brian E Robinson, Yuta J Masuda, Allison Kelly, Margaret B Holland, Charles Bedford, Malcolm Childress, Diana Fletschner, Edward T Game, Chloe Ginsburg, Thea Hilhorst, Steven Lawry, Daniela A Miteva, Jessica Musengezi, Lisa Naughton-Treves, Christoph Nolte, William D Sunderlin, Peter Veit 2018/3 Conservation Letters Volume 11 Issue 2 Pages e12383

PFUFF ET AL., INCLUDING THE BRAZILIAN AMAZON

Pfaff, A., J. Robalino, E. Lima, C. Sandoval, L.D. Herrera (2013). "Governance, Location and Avoided Deforestation from Protected Areas: greater restrictions can have lower impact, due to differences in location". World Dev.

Pfaff, A., J.A. Robalino, D. Herrera and C. Sandoval (2015). "Protected Areas' Impacts on Brazilian Amazon Deforestation: examining conservation - development interactions to inform planning ". PLOS ONE 10(7).

Pfaff, A., J. Robalino, C. Sandoval and D. Herrera (2015). "Protected area types, strategies and impacts in Brazil's Amazon: public PA strategies do not yield a consistent ranking of PA types by impact". Philosophical Transactions B.

[PA regime as the level of government] Herrera D., A. Pfaff and J. Robalino (2019). "Impacts of protected areas vary with the level of government: comparing avoided deforestation across agencies in the Brazilian Amazon". PNAS.

[comparing PA and other types of tenure] Panlasigui, S. , J. Rico-Straffon, A. Pfaff, J. Swenson, and C.J. Loucks (2018). "Impacts of Certification, Uncertified Concessions, Hunting Zones and Protected Areas on Forest Loss in Cameroon, 2000 to 2013". Biological Conservation 227:160-166.

[comparing PA regime over time, following directly upon your #39] Pfaff, A., F. Santiago-Avila and L. Joppa. (2017). "Evolving Protected-Area Impacts in Mexico: political shifts as suggested by

impact evaluations". *Forests* 8(17).

[if taking up time, shifts in context over time] Haruna, A., A. Pfaff, S. van den Ende, and L. Joppa (2014). "Evolving Protected-Areas Impacts in Panama: impact shifts show that plans require anticipation". *Envir. Research Letters* 9.

SOME OTHER CONSERVATION EVALUATION PAPERS

A. Nelson, K. M. Chomitz, Effectiveness of strict vs. multiple use protected areas in reducing tropical forest fires: A global analysis using matching methods. *PLoS One* 6, e22722 (2011).

P. Ferraro et al., More strictly protected areas are not necessarily more protective: Evidence from Bolivia, Costa Rica, Indonesia and Thailand. *Environ. Res. Lett.* 8, 025011 (2013).

Reviewer #2:

Remarks to the Author:

I think *Nature Communications* should accept this manuscript. The manuscript explores Brazil's important issue – the land tenure regime's effect on deforestation. The authors used an interesting and appropriate methodology to compare different land tenure regimes on deforestation over the period 1985-2018. But I expected some additional discussion – beyond those synthesized in the conclusion and other sections – on the political aspects of the deforestation dynamic in the country, particularly in the Amazon region. Also, some recommendations would be very welcome. For example, which kind of land-tenures regime should be recommended to decision-makers as the best option to reduce deforestation in the country— even considering the ambiguities quoted by the authors. Some recommendations in this direction would be helpful if we believe that 95% of deforestation in Brazil is illegal – the point here is: how the lack of law enforcement or illegality in Brazil could be connected to the land tenure regimes affecting deforestation?

Another aspect that the authors could explore is related to the dramatic change in the deforestation profile, particularly in the Amazon region, between 2019 and 2021. In this period, 50% of annual deforestation occurred in public land related to land grabbing (land speculation). A proportion of deforestation was in indigenous and traditional communities' lands. A large proportion (30% of annual deforestation) was in the undesignated public forest (public land). (see the refs. list at the end – it could be helpful).

I was wondering how the finding provided by the authors would change if the analysis had included this period 2019-2021.

In addition, I expected some more discussion on how the distinct rules presented in the Forest Code (the law that protects native vegetation in the different biomes) could affect the analyses, results, and conclusion. For example, the protection by law of native vegetation in the Cerrado biome (Brazilian savanna) is weak compared to the Amazon biome. How could it affect the results?

In conclusion, I consider *Nature Communications* should accept the manuscript if the authors can add more information and discussion about the points raised above.

List of useful references

Kruid, S., Macedo, M. N., Gorelik, S. R., Walker, W., Moutinho, P., Brando, P. M., Castanho, A., Alencar, A., Baccini, A.,; Coe, M. T. (2021). Beyond Deforestation: Carbon Emissions From Land Grabbing and Forest Degradation in the Brazilian Amazon. *Frontiers in Forests and Global Change* 4. <https://doi.org/10.3389/ffgc.2021.645282>

Stabile, M. C. C., Garcia, A. S., Salomão, C. S. C., Bush, G., Guimarães, A. L.,; Moutinho, P. (2022). Slowing Deforestation in the Brazilian Amazon: Avoiding Legal Deforestation by Compensating Farmers and Ranchers. *Frontiers in Forests and Global Change* 4. <https://doi.org/10.3389/ffgc.2021.635638>

Walker, W. S., Gorelik, S. R., Baccini, A., Aragon-Osejo, J. L., Josse, C., Meyer, C., Macedo, M. N., Augusto, C., Rios, S., Katan, T., de Souza, A. A., Cuellar, S., Llanos, A., Zager, I., Mirabal, G. D., Solvik, K. K., Farina, M. K., Moutinho, P.,; Schwartzman, S. (2020). The role of forest conversion, degradation, and disturbance in the carbon dynamics of Amazon indigenous territories and protected areas. *Proceedings of the National Academy of Sciences of the United States of America* 117(6). <https://doi.org/10.1073/pnas.1913321117>

Alencar, A., Castro, I., Laureto L., Guyot, C. Stabile, M., and Moutinho, P. Amazon on Fire - deforestation and fire in undesignated public forests: technical note nº 7. Brasília, DF: Amazon Environmental Research Institute, 2021. Available at: <https://ipam.org.br/bibliotecas/amazon-on-fire-deforestationand-fire-in-undesignated-public-forests/>.

Lapola, D. M., Martinelli, L. A., Peres, C. A., Ometto, J. P. H. B., Ferreira, M. E., Nobre, C. A., Aguiar, A. P. D., Bustamante, M. M. C., Cardoso, M. F., Costa, M. H., Joly, C. A., Leite, C. C., Moutinho, P., Sampaio, G., Strassburg, B. B. N.,; Vieira, I. C. G. (2014). Pervasive transition of the Brazilian land-use system. *Nature Climate Change*, 4(1). <https://doi.org/10.1038/nclimate2056>

Two additional comments

Considering the authors' focus on Brazil, I am not sure if the title (tropical deforestation) is appropriate. Consider change it.

Replace the ref. #35 by Azevedo-Ramos, C., Moutinho, P., Arruda, V. L. da S., Stabile, M. C. C., Alencar, A., Castro, I; Ribeiro, J. P. (2020). Lawless land in no man's land: The undesignated public forests in the Brazilian Amazon. *Land Use Policy* (January), 104863. <https://doi.org/10.1016/j.landusepol.2020.104863>.

Reviewer #1 (Remarks to the Author):

Comments

This manuscript takes up a very important question. It also provides quite a lot of solid analyses. I would say, certainly, that this work (and prior works) support the importance of tenure regimes. Also, I definitely feel highlighting undesignated+untitled ('UnUn') lands is a useful policy guide. Consequently, I feel very confident that, in its final version, this effort will contribute quite a lot. That said, I'd also suggest multiple significant additions are required to arrive at its final version. Additions are needed not due to any mistake but rather just challenges inherent in a big question; thus, with only positive intentions I offer some ideas below on refinements and communications.

Response:

We thank this reviewer for this positive assessment and for the thoughtful suggestions for further improving the clarity of key messages.

Contextualization Most confusing for me, despite all the well-motivated and significant effort to distinguish then consider separately quite a number of settings, was perhaps under-publicizing a 'flip' in results for Amazonia (which seemingly is where a lot of these UnUn lands are located?). While I'd agree highlighting the lack of governance structures for UnUn is a really helpful thing, leaning towards single conclusions without conditioning upon contexts that differ enormously in terms of the reach / capacity of public governance – for any given tenure – was confusing for me. (Specifically, e.g., is a protected area in Sao Paulo usually very similar to PAs within Rondonia?)

Having gotten through the paper, I noticed more in the Abstract the “except in remote ...” clause which qualifies the prior statements. [...]

Response:

We fully agree that our results for undesignated/untitled public lands in the Amazonian context are particularly interesting and relevant, and we happily follow this reviewer's suggestion to more clearly highlight the 'flip' in Amazonian results for private vis-à-vis undesignated/untitled regimes, including in the abstract. We also fully agree that protected areas in São Paulo and Rondônia are commonly very different in terms of the reach/capacity of public governance (especially concerning enforcement of existing laws). We have made several text additions to expand the regional contextualization and make these differences more explicit, which indeed very plausibly contribute to explaining Amazonia's 'flip' in results. Specifically:

- 1) We now include the explicit wording “except in Amazonia where on-the-ground governance can be limited...” in the abstract, replacing the previous wording that had merely indirectly hinted at Amazonia by characterizing its remote/low-governance context. Moreover, we now emphasize Amazonia's special case in the section subtitle where Amazonian effects are explained in detail (lines 234-291 in the main manuscript).
- 2) We added the sentence “This lower capacity for law enforcement in public areas, in turn, may reduce the difference in the *de facto* governance of Amazonia's undesignated/untitled lands vs. its conservation/indigenous lands (relative to differences in other biomes).” (lines 256-258)

[...] Yet in way, I guess I might summarize that “poorly defined” to some extent effectively describes what one was supposed to take away from the Abstract with respect to “well-defined” non-private regimes. Put another way, I think the paper is broadly right to want to include multiple elements in “regime”, yet maybe political-economic factors add too? Along these lines, on the positive side I am glad to see Table 1 include stories in either direction. Allowing for more than one empirical conclusion also could be okay, if each fits a given context (by which I mean not each of 50+ combinations but, say, “the governed versus the far frontier”?).

Response:

We'd like to clarify that with our choice of the terms "poorly defined tenure" vs. "well-defined tenure", we directly adopt terminology that is commonly used in the context of land tenure in fields such as Development Studies or Resource Economics, to specifically describe how well each of the dimensions of rights and regulations that jointly constitute land tenure is *legally* defined. To improve clarity in this respect in the Abstract, we have rephrased the corresponding sentence to: "We find that public tenure regimes with poorly defined tenure rights increased deforestation relative to any alternative regime in most contexts"

We agree that various socio-political and socio-economic factors beyond land tenure causally affect deforestation rates. Thus, if one follows a broad definition of "governance" that goes beyond regulatory factors, one could certainly argue that such factors contribute to "*de facto* governing" forests. However, our study's specific focus is on specifically *land-tenure-related* governance factors (emphasis added). We defined "land-tenure regimes" accordingly, to include the bundles of rights and regulations that collectively define land tenure (see **Table S2**), including whether/which specific land-related policies might apply as a direct result of the land-tenure form in place (see lines **12-17**).

While we thus would not include political or economic factors into our definition of land tenure regimes, we do acknowledge their effects, and we aim to account for them by including proxies for such factors as matching and modeling covariates, and focusing on effects that emerge as generalizable across different contexts characterized by different political and socioeconomic situations. Disentangling the many non-regulatory (as well as regulatory) factors that determine forest "governance" *sensu lato*, however, is far beyond our intended study scope, and would require very different study designs.

Similarly, our study's principal aim is to establish to what extent effects are consistent across diverse socio-environmental contexts. Characterizing the specific factors that define the bounds for each effect's consistency, in turn, is beyond the scope of our study. It would certainly be very interesting to explore how certain factors (e.g., including whether or not a region is "far frontier") cause our assessed land-tenure effects to vary in space and time, but this would require (probably several) follow-up studies, again, with devoted study designs.

Nevertheless, we fully agree with this reviewer that further research into the specific context-distinguishing factors would be much-needed next steps, especially for those effects that showed only limited consistency across different contexts, which is why we explicitly call for such studies in our conclusions (lines **319-322**: "For those tenure regimes for which our assessment does not indicate high generalizability or consistency of effects across scales, such as IPLC-based regimes, guidance to sustainability policies should be based on further research into the context-distinguishing factors").

Standardization It is an important point to highlight for readers that matching drops some areas; quite intentionally, in fact, as one endeavors to find control observations in the same contexts as those in which the treatments are found. Thus, e.g., the ATT and ATU impacts could vary, as if treatments are in particular spaces, while impacts differ across spaces, ATT does not generalize. Nonetheless, despite the utility from awareness and communication of these facts, I feel it would be much more direct and clear to simply provide the "In The Context We Found Them" estimate of the impacts of these various regimes. It is for those places that we can really estimate impacts.

Naturally, there are good reasons for standardization, especially for the authors' goals of ranking across regimes/interventions in terms of impact. So the weighted results could also be of interest. Yet I do not feel the average readers will really have any idea about what they are looking at here. Since I think the weighting has some fine justifications and is being done in okay and transparent manner, for those who bother to look, I use the term "addition" for adding unweighted estimates. What might add further, then, is highlighting when and why weighted conclusions have differed?

Being more specific, really I'd lead with the Where They Were Found estimates of these impacts. Further, I'd emphasize that non-random sitings probably tell us something about the net benefits. Then if the authors think rankings for outside those sites are useful, explain when rankings shift?

Response:

We agree that matching involves dropping unmatched observations, and that this is important for readers to appreciate, which is why we had clearly described this in our methods section (lines 410-413 in main manuscript: "We first applied coarsened-exact matching implemented in the 'cem' package (57) in R (versions 3.5.1-4.0.2) (58), which involves temporarily 'coarsening' each confounding variable into bins (predetermined strata), and dropping unmatched observations from the sample". As suggested, we have further clarified this fact in the main text ("After dropping the unmatched land parcels, we assessed in how far effects estimated with the remaining, matched data subsets would be generalizable to the entire population of land parcels, using Tipton's index of generalizability..." lines 80-82 in main manuscript).

However, we would like to clarify that we subsequently estimated the generalized effects for the entire population of land parcels, precisely to be able to extend our statements beyond the specific contexts of the parcels in the matched data subsets. This generalization involved multivariate regression on the matched parcels, but in a way that each parcel's contribution to the estimated effects was weighted such that the distribution of all matched parcels taken together in multivariate covariate space would closely approximate that of the entire population of parcels. In terms of our quasi-experimental study design, we thereby combine the advantage of classical matching noted by this reviewer (i.e., finding control observations in the same contexts as those in which the treatments are found) with the capacity of (weighted) regression to generalize to larger populations (provided that generalizability is, indeed, found).

Our assessment of the generalizability, and the subsequent weighing of effects, was precisely to better understand where ATT (rather, in our case, ATM) does or does not generalize to the total population of land parcels at the respective spatiotemporal scale. We aimed to communicate as clearly as possible what these effect estimates mean in the context of the scale-specific "entire populations" of land parcels. However, we very much appreciate the reviewer's concerns about the intuitive understandability of these results for the average reader. Having carefully reconsidered both the advantages and the disadvantages of presenting the unweighted results alongside the weighted results in the main manuscript, we arrived at the following solution (in the interest of minimizing possible misunderstandings of our results):

- 1) We favored primarily presenting the weighted results in the main manuscript, because we consider the indirect comparisons across tenure regimes (which require using weighed results for fair cross-comparisons) as critical components of our paper, and indeed as key conceptual advances over previous knowledge on land-tenure effects on deforestation that was mainly based on (non-comparable) studies analyzing single, or few tenure forms at a time. This primacy of the weighted results is also consistent with our primary intention of providing synthesis of effects across different tenure regimes and contexts (i.e., rather than informing about any particular tenure regime's specific effects under certain very specific contexts).
- 2) We did consider combining both sets of weighted and unweighted results in the main paper and tested the implications of such decision for a few paragraphs. However, we found that this would require trying to convey a much more complex story, likely making it substantially more difficult for many readers to identify the most important messages in each paragraph.
- 3) We do completely agree with this reviewer on the importance of highlighting cases where our weighted effect estimates do/do not qualitatively differ from the

unweighted ones. Generally, the weighed effect estimates were indeed most often qualitatively very similar to the unweighted one. However, to make our readers aware of where there were differences/no differences in these estimates (i.e. where effects were generalizable/non-generalizable to the respectively entire population of parcels), we communicate this throughout the text in the following ways:

- a. We report where effects were generalizable to the entire population of land parcels, as exemplified here by lines **105-109** in main manuscript: “Tipton’s generalizability index also indicated that covariate distributions were similar across all alternative tenure-regime comparisons and undesignated/untitled land parcels, meaning these results are highly generalizable to the entire population of land parcels at the respective spatial-temporal scales (see **SI Appendix; Fig. S3/S4, Tables S3/S6-7**)”. See also lines **198-200** for similar statements on the generalizability of effect estimates for conservation regimes, and lines **247-248** for the generalizability of effects estimates for private and undesignated/untitled regimes in Amazonia.
- b. Conversely, we now more explicitly discuss where effects were not generalizable to the entire population of land parcels. Note that the only tenure-regime comparisons for which the weighted effect estimates substantially differed from the unweighted estimates were those for the effects of indigenous regimes vis-à-vis private regimes, with the unweighted estimates overestimating the deforestation-decreasing effects of indigenous lands. This can be explained by their biased siting in all biomes except in the Cerrado. Given the associated limited generalizability from matched parcels to the entire parcel populations (in the other biomes than the Cerrado), this could imply that the ambiguity of these effects could in reality be even greater. We changed two sentences (lines **218-223** in the main manuscript) to make this clearer. They now read: “Indigenous tenure reduced deforestation vis-à-vis private tenure in only 59.4-70.4% of cases, although true population-wide effects might be even more ambiguous. This is because these latter effects were only generalizable from matched parcels to larger parcel populations in 17% of cases, mostly in the Cerrado biome (T-index ≥ 0.5 in **Table S4**), reflecting the biased siting of indigenous reserves in other biomes, specifically in areas farther from cities and at higher elevations, relative to the population averages (**Table S10**).”
- c. In the methods section, we have added clarifying text on the overall covariate differences in the land parcels that were matched and the entire population of land parcels (i.e., the latter including the unmatched land parcels that were dropped). Specifically, we now detail what the calculation of the absolute standardized mean difference (ASMD) of each covariate indicates for matched and entire populations of land parcels (results of ASMD calculations are found in the SI Appendix, Table **S10**): “These ASMD indicated that matched data subsets were, on average, at lower elevations, farther away from cities, and of larger areas than the average land parcel in Brazil. We found no systematic differences in these patterns across spatial or temporal scales (SI Appendix, Table S10).” (lines **440-444** in main manuscript)

Connection For putting the manuscript’s and the above additional points in perspective, I suspect perhaps some additional references to, and connections, to prior literatures could add for readers:

- I’m not in tenure literatures, per se, but my impression is many have been, for some time; for instance, I’ll mention William Sunderlin, who seems uncited while it took me no time to find relevant works [listed below & highly cited, even if maybe not his best fits here?]

- I am in impact evaluation, which has been growing for deforestation – but perhaps less in ways relevant for the conservation-policy comparisons here? Yet grabbing just one author cited here, again it took little time to find comparison of regimes (surely there are more!?)

Response:

It was indeed important for us to connect our study's findings to existing research, but given that the literature on land tenure is very diverse and spans different scientific fields (as seen in our overview of hypothesized mechanisms, **Table S1**), we may have missed some key references so we are very grateful for these pointers. We had indeed already made some of the suggested connections to prior literature. For example, maybe due to the required citation style, it might have appeared as if William Sunderlin was not cited in our paper, although two articles by Sunderlin had already been cited:

- Ref #11 in main manuscript: Tseng, T.-W. J., Robinson, B. E., Bellemare, M. F., BenYishay, A., Blackman, A., Boucher, T., Childress, M., Holland, M. B., Kroeger, T., Linkow, B., Diop, M., Naughton, L., Rudel, T., Sanjak, J., Shyamsundar, P., Veit, P., **Sunderlin, W.**, Zhang, W., & Masuda, Y. J. (2020). Influence of land tenure interventions on human well-being and environmental outcomes. *Nature Sustainability*, 1–10. <https://doi.org/10.1038/s41893-020-00648-5>.
- Ref #10 in Supplementary Information (SI): Robinson, B. E., Masuda, Y. J., Kelly, A., Holland, M. B., Bedford, C., Childress, M., Fletschner, D., Game, E. T., Ginsburg, C., Hilhorst, T., Lawry, S., Miteva, D. A., Musengezi, J., Naughton-Treves, L., Nolte, C., **Sunderlin, W. D.**, & Veit, P. (2017). Incorporating Land Tenure Security into Conservation: Conservation and land tenure security. *Conservation Letters*, 11(2), e12383. <https://doi.org/10.1111/conl.12383>.

We had chosen those two as they provide broad reviews of the land tenure/deforestation-related literature up to this date.

Following suggestions from both reviewers, we added several more key references in order to better connect our findings to the most relevant tenure literature. Specifically, we added the following citations which *i*) better contextualize how land tenure is a central challenge for decreasing tropical deforestation rates (Larson 2013), *ii*) better relate our findings to deforestation in undesignated/untitled lands in Brazil (Yanai 2022), and *iii*) better recognize current understandings on the factors that drive protected area effectiveness in the tropics (Haruna 2014 and Ferraro 2013):

- Ref. #4 Larson, A. M., Brockhaus, M., **Sunderlin, W. D.**, Duchelle, A., Babon, A., Dokken, T., Pham, T. T., Resosudarmo, I. A. P., Selaya, G., Awono, A., & Huynh, T.-B. (2013). Land tenure and REDD+: The good, the bad and the ugly. *Global Environmental Change*, 23(3), 678–689. <https://doi.org/10.1016/j.gloenvcha.2013.02.014>
- Ref. #37 A. M. Yanai, P. M. L. de Alencastro Graça, L. G. Ziccardi, M. I. S. Escada, P. M. Fearnside, Brazil's Amazonian deforestation: the role of landholdings in undesignated public lands. *Reg Environ Change* **22**, 30 (2022)
- Ref. #41 Haruna, A., A. Pfaff, S. van den Ende, and L. Joppa (2014). "Evolving Protected-Areas Impacts in Panama: impact shifts show that plans require anticipation". *Envir. Research Letters* 9.
- Ref. #45 P. Ferraro et al., More strictly protected areas are not necessarily more protective: Evidence from Bolivia, Costa Rica, Indonesia and Thailand. *Environ. Res. Lett.* 8, 025011 (2013).

The reason for connections with literature on tenure and conservation regimes is not only that the past work offers context. Also, a relatively simple extension now becoming more common might contribute. I think that the matching done here is perfectly reasonable for what it is, i.e., some use of fixed characteristics (elevation, slope, travel time), at least for the most part. That

surely helps, very much in the spirit of, say, the early Lucas Joppa work on locational bias at even larger scale. However, I think we know now that such matching certainly does not guarantee “parallel trends”, while simply extending such matching to pre-treatment deforestation outcomes can help with it. For comparing such different contexts, I’d suspect it might be very helpful to see if this matters.

Response:

We appreciate this comment, as we are well aware of the possible bias, and thus address it carefully in the supplementary information. As noted therein (lines 574-615 in SI), we acknowledge that the estimated effects of tenure-regime differences could in principle have been affected by differences in initial forest cover between our matched parcels that resulted from forest-to-agriculture conversions prior to the respective treatment periods. Hence, we assessed possible bias in our conclusions due to systematic differences in initial forest cover. We modelled the initially forest-covered percentages of the matched parcels’ areas at each spatiotemporal scale as a function of their treatment (i.e., tenure-regime identity), and detected no systematic unidirectional differences between treatment and control across scales, indicating that our main conclusions are not biased by such differences (see Fig S6). However, we found differences in either direction in individual cases and thus cannot rule out that these might partly explain differential forest trajectories for some tenure regimes and spatiotemporal scales. We addressed this caveat by basing our main conclusions on results that showed consistency across spatiotemporal scales and by ruling out this bias when drawing insights from scale-specific results (e.g., the changing relative effectiveness of tenure regimes in curbing Amazonian deforestation).

We chose this indirect approach over directly matching parcels on initial forest cover. This was motivated, firstly, by our aim to evaluate all tenure regimes via a consistent modelling protocol. Here, retaining sufficient degrees of freedom for each tenure regime and spatiotemporal scale required us to constrain the total number of matching covariates, as that number affects both the matched dataset sizes and the number of modelling covariates included in the binomial GLMs. Secondly, our specific aim was not to assess total forest losses of different tenure regimes over their entire lifetimes (which would necessitate accounting for any prior deforestation already internalized in parcels’ initial forest cover), but to assess whether tenure regimes consistently differed in their ability to retain remaining forest cover over different time periods (defined by their unique historical deforestation trends, policies, etc.). Here, differences in the magnitude of additional percentage losses among the matched parcels are already internalized in the way percentages are modelled by binomial GLMs. Finally, parcel-level differences in initial forest cover do not necessarily reflect prior forest-to-agriculture conversions, but may also reflect natural spatiotemporal heterogeneity in land cover (e.g., due to mosaics of forest and non-forest vegetation, landslides, etc.) as well as earlier agricultural expansion over non-forest vegetation, particularly outside the Amazonia biome.

We address specific references below:

Two Quick Literatures Examples (Sunderlin for tenure regimes and their consequences, Pfaff for conservation regimes’ impacts)

SUNDERLIN QUICK LOOK HERE → MAYBE COULD ADD SOME OTHER LONGSTANDING TENURE FOLKS TOO?

From exclusion to ownership? Challenges and opportunities in advancing forest tenure reform. William D Sunderlin, Jeffrey Hatcher, Megan Liddle 2008 Rights and Resources Initiative

The content of this report from RRI is well-represented in the review by Tseng et al. 2020 highlighted above (ref. #11 in main manuscript)

Land tenure and REDD+: The good, the bad and the ugly. AM Larson, M Brockhaus, WD Sunderlin, A Duchelle... - Global environmental change, 2013

We have added this citation (line 7 in main manuscript, ref. #4).

Linking forest tenure reform, environmental compliance, and incentives: lessons from REDD+ initiatives in the Brazilian Amazon. AE Duchelle, M Cromberg, MF Gebara, R Guerra... - World Development, 2014

Indeed, we had already cited this paper (ref #11 in the SI).

Incorporating land tenure security into conservation. Brian E Robinson, Yuta J Masuda, Allison Kelly, Margaret B Holland, Charles Bedford, Malcolm Childress, Diana Fletschner, Edward T Game, Chloe Ginsburg, Thea Hilhorst, Steven Lawry, Daniela A Miteva, Jessica Musengezi, Lisa Naughton-Treves, Christoph Nolte, William D Sunderlin, Peter Veit 2018/3 Conservation Letters Volume 11 Issue 2 Pages e12383

Indeed, we had already cited this paper (ref #10 in the SI).

PFaff ET AL., INCLUDING THE BRAZILIAN AMAZON

Pfaff, A., J. Robalino, E. Lima, C. Sandoval, L.D. Herrera (2013). "Governance, Location and Avoided Deforestation from Protected Areas: greater restrictions can have lower impact, due to differences in location". World Dev.

Pfaff, A., J.A. Robalino, D. Herrera and C. Sandoval (2015). "Protected Areas' Impacts on Brazilian Amazon Deforestation: examining conservation - development interactions to inform planning". PLOS ONE 10(7).

Instead of citing the above 2013 and 2015 papers, we have cited Herrera et al. 2019 (below) due to its relevance to the entire Amazon (not only Acre), and its overarching summary of the impacts of PAs in Brazil.

Pfaff, A., J. Robalino, C. Sandoval and D. Herrera (2015). "Protected area types, strategies and impacts in Brazil's Amazon: public PA strategies do not yield a consistent ranking of PA types by impact". Philosophical Transactions B.

Already cited (ref #47 in the SI).

[PA regime as the level of government] Herrera D., A. Pfaff and J. Robalino (2019). "Impacts of protected areas vary with the level of government: comparing avoided deforestation across agencies in the Brazilian Amazon". PNAS.

Already cited (ref. #34 in the SI).

[comparing PA and other types of tenure] Panlasigui, S. , J. Rico-Straffon, A. Pfaff, J. Swenson, and C.J. Loucks (2018). "Impacts of Certification, Uncertified Concessions, Hunting Zones and Protected Areas on Forest Loss in Cameroon, 2000 to 2013". Biological Conservation 227:160-166.

This is an evaluation of the Forest Stewardship Council's certification schemes in decreasing deforestation in Cameroon, and is thus pertinent to market-based instruments for the reduction of deforestation, rather than to the regulation of property rights via different tenure regimes.

[comparing PA regime over time, following directly upon your #39] Pfaff, A., F. Santiago-Avila and L. Joppa. (2017). "Evolving Protected-Area Impacts in Mexico: political shifts as suggested by impact evaluations". Forests 8(17).

[if taking up time, shifts in context over time] Haruna, A., A. Pfaff, S. van den Ende, and L. Joppa (2014). "Evolving Protected-Areas Impacts in Panama: impact shifts show that plans require anticipation". Envir. Research Letters 9.

We have added Haruna et al. 2014 as a citation (lines 183-184 in main text, ref #41) to exemplify evidence on (spatial and temporal) inconsistencies in the effectiveness of conservation-focused regimes.

SOME OTHER CONSERVATION EVALUATION PAPERS

A. Nelson, K. M. Chomitz, Effectiveness of strict vs. multiple use protected areas in reducing tropical forest fires: A global analysis using matching methods. PLoS One 6, e22722 (2011).

Already cited (ref. #42 in main manuscript)

P. Ferraro et al., More strictly protected areas are not necessarily more protective: Evidence from Bolivia, Costa Rica, Indonesia and Thailand. Environ. Res. Lett. 8, 025011 (2013).

We have added this citation in line 201 in the main manuscript, to relate to our finding that sustainable use areas may most reliably achieve *large* reductions in deforestation: "This aligns with findings from other tropical countries that stricter protection does not necessarily increase conservation outcomes (45)."

Reviewer #2 (Remarks to the Author):

I think Nature Communications should accept this manuscript. The manuscript explores Brazil's important issue – the land tenure regime's effect on deforestation. The authors used an interesting and appropriate methodology to compare different land tenure regimes on deforestation over the period 1985-2018.

Response:

We thank this reviewer for this positive assessment and the thoughtful comments.

But I expected some additional discussion – beyond those synthesized in the conclusion and other sections – on the political aspects of the deforestation dynamic in the country, particularly in the Amazon region. [...]

Response:

We designed our cross-contextual study to cover multiple time periods under very different political regimes, precisely to identify effects that hold generally, i.e., beyond particular political contexts. Our study design thus does not permit evaluating the effectiveness of any specific policies on deforestation rates in different land tenure regimes, nor the effects of any specific political factors. Some particularly important variations in policies between Amazonia and other biomes, particularly in the Forest Code and in commodity moratoria, however, are captured via the different spatiotemporal scales across which we repeated our tests. Since we cannot directly attribute effects to specific policies nor particular political events, we wish to remain careful to only highlight those links that we deem sufficiently supported by our results.

We have added and edited text to make those specific links clearer than in the previous version. In particular:

- 1) we now more clearly address the current political situation with respect to Amazonia in several sections, referring particularly to plans by Brazil's current administration to privatize Amazonian indigenous and protected areas, In particular, in lines **289-291**, we now write “Informing current political debates, these insights indicate that the privatization of protected or indigenous lands would likely increase deforestation in Amazonia, or elsewhere (16, 52)”, replacing previous text that merely referred to such political debates.
- 2) we now more explicitly refer to several specific policies implemented under previous administrations, specifically the Forest Code with its differential forest-cover requirements and the soy and beef moratoria, as key policies that likely contributed to explaining our results for Brazil, and for Amazonia, in particular (lines **252-253** in main text). This replaces earlier text that only generically referred to commodity moratoria and policies requiring higher forest cover in Amazonia. Later in this paragraph we now more explicitly state that our findings indicate that these similarly strict private-actor-targeting policies measures may have the potential to be successful in reducing deforestation rates in other biomes: “Moreover, this suggests that there is potential to substantially decrease future Brazil-wide deforestation rates by extending more stringent private-actor-focused environmental policies (e.g., as currently via the Forest Code for Amazonia), to Brazil's other remote biomes, where remaining forestlands are mostly private (Cerrado: 80.4%; Pantanal: 92.8%; Fig. 1b-c)”. (lines **263-266**)

We additionally extended our discussion of implications of other results:

- In lines **111-112**, we reworded to “policy interventions to install alternative tenure regimes” [in undesignated/untitled lands], now cite Azevedo-Ramos et al 2020, replacing the previous 2018 citation as suggested, and we additionally cite Yanai et. al. 2022 (Regional Environmental Change; ref. #37) to support our statement on appeals for policy interventions in these lands.

- In lines **127-136**, we had already provided background to how different policies have promoted the privatization/formalization of undesignated/untitled lands, and related these arguments to our results in lines **144-149**. In lines **152-154**, we have added the phrase “highlighting the importance of coupling titling interventions with environmental policies to effectively safeguard forests”, to more clearly link our results to Probst et al.’s (2020) findings on titling interventions.

[...] Also, some recommendations would be very welcome. For example, which kind of land-tenures regime should be recommended to decision-makers as the best option to reduce deforestation in the country— even considering the ambiguities quoted by the authors. [...] Some recommendations in this direction would be helpful if we believe that 95% of deforestation in Brazil is illegal – the point here is: how the lack of law enforcement or illegality in Brazil could be connected to the land tenure regimes affecting deforestation?

Response:

As we state in our Conclusions section (lines **300-302**, ff.), our study offers “guidance to policy-makers about which of alternative tenure interventions might reduce long-term deforestation rates most effectively and most reliably across different socio-environmental settings” – however, our aim is not to be prescriptive, so we purposefully avoided giving concrete recommendations for any specific tenure policies or interventions. We believe that our section in lines **310-322** provides rather clear statements on what the likely implications for specifically deforestation rates are for several policy measures that decision-makers may or may not consider. However, since land-tenure interventions will likely affect not just deforestation rates, but also various other societal goals that decision-makers will need to consider (e.g., related to human rights, livelihoods, etc., with potentially large trade-offs), and given that such other goals were not part of our analyses, we would prefer to retain our roles to that of honest brokers in this case. We hope that this reviewer will find our position acceptable.

Regarding the second point regarding how the lack of law enforcement in Brazil relates to land-tenure regimes affecting deforestation, we cannot provide a simple answer. The fact that private regimes perform comparatively better in Amazonia suggests that legal instruments targeting private actors (e.g., the Forest Code) might not be completely ineffective, after all – despite the region’s remoteness and its (real or perceived) generally low stringency of environmental law enforcement. Or at least, our results support the notion that in a hypothetical alternative world in which such legal instruments had only been as mild in Amazonia as in other biomes, Amazonian deforestation rates would have been even higher. However well- or ill-functioning law enforcement in Brazil might be, our analyses suggest that these levels of enforcement apparently suffice to make the existing private-actor-focused laws work at least in the direction they should. Of course, the real question this reviewer cares about might not be whether they work *at all*, but how effective they are, and whether stronger law enforcement might not make them *more effective*. However, given that our study was not designed to directly analyze how differential law enforcement levels might have caused variations in land-tenure effects, we, cannot make strong statements on this.

To nevertheless acknowledge the plausibly very important role of law enforcement, we complemented our existing discussion sentences on this with the following:

“This lower capacity for law enforcement in public areas, in turn, may reduce the difference in the *de facto* governance of Amazonia’s undesignated/untitled lands vs. its conservation/indigenous lands (relative to differences in other biomes).” (lines **256-258**)

Another aspect that the authors could explore is related to the dramatic change in the deforestation profile, particularly in the Amazon region, between 2019 and 2021. In this period, 50% of annual deforestation occurred in public land related to land grabbing (land speculation).

A proportion of deforestation was in indigenous and traditional communities' lands. A large proportion (30% of annual deforestation) was in the undesignated public forest (public land). (see the refs. list at the end – it could be helpful).

I was wondering how the finding provided by the authors would change if the analysis had included this period 2019-2021.

Response:

Indeed, data from INPE and Mapbiomas indicate that deforestation rates have started increasing since 2017. At the same time, Brazil's current administration clearly marks a new epoch in terms of forest policies, regulations, monitoring and enforcement, as well as in overall governance and management incentives, that contrasts with the earlier epochs that we analyzed in this study.

We recognize that several sources have reported that recent deforestation has predominantly taken place in undesignated public lands (Kruid et al. 2021, IPAM technical report). Indigenous lands, in fact, are not reported therein as having increasing deforestation rates. To better link our results to these recent deforestation increases on undesignated public lands, we have added text that more clearly links to this new epoch of forest governance. In particular, we now refer to a study finding that most of this recent deforestation has occurred in undesignated/untitled lands, while highlighting the importance of interventions on these lands: “[...] As such, they [our findings] substantiate appeals for policy interventions to install alternative tenure regimes (36, 37), which may be particularly urgent given increasing deforestation rates in these lands observed since 2017 (38)” (lines 111-113). Additionally, we now better link to recently highlighted deforestation risks for Amazonian indigenous lands, by more explicitly laying out our study's implications for currently debated policies to privatize such lands: “Informing current political debates, these insights indicate that the privatization of protected or indigenous lands would likely increase deforestation in Amazonia, or elsewhere.(16, 52)” (lines 289-291).

Although we agree that, in principle, it would be interesting to study how different tenure regimes may have influenced these recent deforestation increases specifically in this new political epoch, we feel that studying the current administration's impacts on land-tenure-deforestation effects cannot and should not form part of this study, for three main reasons:

- 1) Firstly, options to extend our study until 2021 are severely limited by the timeliness of available data. In particular, the population density variable (which is key for our matching and regression analyses) would need to be extrapolated from 2015, with every additional year after 2015 becoming much less reliable. To approximate average conditions of this variable in our study's final 2012-2018 epoch, it seemed sensible to make the short-term extrapolation until 2018 (given that the epoch centers around 2014/15 and the mean values over the period are still mostly data-driven). However, we would not be comfortable relying *purely* on extrapolated values (and extrapolated for *more years*) for a 2019-2021 period. Similarly, even the most recently published update of the MapBiomas land-cover data only goes until 2020, and while reliably extrapolating land-cover into later years via models is generally tricky, it would in fact prohibit rigorous causal inference on the drivers of the modelled patterns (as we would merely learn back the assumptions used for our extrapolation).
- 2) Secondly, even if more timely data were available, we would be concerned by adding only a two-year slice (2019-2020) cut off from a longer governance epoch, and particularly one that is still ongoing, as this would be an undesirable break in our otherwise consistent study design, which compares effects retrospectively across multiple (completed) epochs.

- 3) Finally, we purposefully designed our study to focus on “long-term deforestation impacts of land-tenure regimes”. Besides our main motivation for this design choice (i.e., to synthesize in how far the effects are generalize beyond particular contexts), another key reason for this was to avoid biases due to short-term “clear-to-claim” processes. Such short-term “clear-to-claim” processes (i.e., rapid clearing of land in speculation of attaining greater land rights or of lowering economic losses due to anticipated future regulations) could interfere with our analysis of directional causal effects of land tenure on forest cover. This is also why we did not consider forest-cover changes on a yearly basis but only aggregate changes over longer, multi-year epochs, and also why we excluded land parcels under the Terra Legal program. Thus, precisely some of those recent, presumably speculation-driven deforestation spikes that this reviewer mentioned could in fact make a temporal extension of our study of just a few years substantially less robust. We believe that such effects are more appropriately explored via studies designed to capture short-term impacts of specific land-tenure-change events (e.g., such as in the recent study of Amazonian land titling by Probst et al.; #13 of our references), rather than to capture longer-term effects of land-tenure regimes.

In addition, I expected some more discussion on how the distinct rules presented in the Forest Code (the law that protects native vegetation in the different biomes) could affect the analyses, results, and conclusion. For example, the protection by law of native vegetation in the Cerrado biome (Brazilian savanna) is weak compared to the Amazon biome. How could it affect the results?

Response:

Differences in the stringency of the Forest Code regulations in Amazonia vs. other biomes would not actually “affect” our results, as they are indeed implicit in our study design (i.e., defining tenure regimes as including predisposition to policies, comparing effects across biomes, and highlighting notably exceptions to “general” effects).

However, we fully agree that the role of the regionally different Forest Code regulations could have been discussed more explicitly, which we addressed with two changes:

- 1) First, we now more explicitly name the Forest Code as a plausible key factor behind contrasting results for private-tenure effects on deforestation for Amazonia vs. other biomes (thereby replacing previous text where we had only generically referred to the stringency of policies): “These counter-intuitive Amazonian effects might be explained by the region’s specific environmental governance setting. Over recent decades, Amazonian private landholders have been subject to stricter forest-protection policies than those in other biomes, including the Forest Code’s requirement to retain four times the forest cover required in other biomes, as well as earlier-implemented soy and beef moratoria (47, 48)” (lines 249-253).
- 2) Second, we now more explicitly suggest that more stringent requirements under the Forest Code could be effective in reducing deforestation in similarly remote, mostly privately-held biomes: “Moreover, this suggests that there is potential to substantially decrease future Brazil-wide deforestation rates by extending more stringent private-actor-focused environmental policies (e.g., as currently via the Forest Code for Amazonia), to Brazil’s other remote biomes, where remaining forestlands are mostly private (Cerrado: 80.4%; Pantanal: 92.8%; Fig. 1b-c)” (lines 263-267).

In conclusion, I consider Nature Communications should accept the manuscript if the authors can add more information and discussion about the points raised above.

Response:

We thank this reviewer again for the helpful comments.

List of useful references

Kruid, S., Macedo, M. N., Gorelik, S. R., Walker, W., Moutinho, P., Brando, P. M., Castanho, A., Alencar, A., Baccini, A.; Coe, M. T. (2021). Beyond Deforestation: Carbon Emissions From Land Grabbing and Forest Degradation in the Brazilian Amazon. *Frontiers in Forests and Global Change* 4. <https://doi.org/10.3389/ffgc.2021.645282>

Stabile, M. C. C., Garcia, A. S., Salomão, C. S. C., Bush, G., Guimarães, A. L.; Moutinho, P. (2022). Slowing Deforestation in the Brazilian Amazon: Avoiding Legal Deforestation by Compensating Farmers and Ranchers. *Frontiers in Forests and Global Change* 4. <https://doi.org/10.3389/ffgc.2021.635638>

Walker, W. S., Gorelik, S. R., Baccini, A., Aragon-Osejo, J. L., Josse, C., Meyer, C., Macedo, M. N., Augusto, C., Rios, S., Katan, T., de Souza, A. A., Cuellar, S., Llanos, A., Zager, I., Mirabal, G. D., Solvik, K. K., Farina, M. K., Moutinho, P.; Schwartzman, S. (2020). The role of forest conversion, degradation, and disturbance in the carbon dynamics of Amazon indigenous territories and protected areas. *Proceedings of the National Academy of Sciences of the United States of America* 117(6). <https://doi.org/10.1073/pnas.1913321117>

Alencar, A., Castro, I., Laureto L., Guyot, C. Stabile, M., and Moutinho, P. Amazon on Fire - deforestation and fire in undesignated public forests: technical note nº 7. Brasília, DF: Amazon Environmental Research Institute, 2021. Available at: <https://ipam.org.br/bibliotecas/amazon-on-fire-deforestationand-fire-in-undesignated-public-forests/>.

Lapola, D. M., Martinelli, L. A., Peres, C. A., Ometto, J. P. H. B., Ferreira, M. E., Nobre, C. A., Aguiar, A. P. D., Bustamante, M. M. C., Cardoso, M. F., Costa, M. H., Joly, C. A., Leite, C. C., Moutinho, P., Sampaio, G., Strassburg, B. B. N.; Vieira, I. C. G. (2014). Pervasive transition of the Brazilian land-use system. *Nature Climate Change*, 4(1). <https://doi.org/10.1038/nclimate2056>

Response:

Of the above-suggested references, we have added Kruid et al 2021 (ref. #38), which is most closely relevant to our findings, whereas some of the other references are more related to alternative methods for decreasing deforestation (Stable et al. 2022), quantification of Amazonian forest degradation (Walker et al. 2020), or implications for carbon (Lapola 2014). We have additionally added a reference to Yanai et al. 2022 (Regional Environmental Change) (ref. #37), to better relate our findings to observed deforestation in undesignated/untitled lands.

Two additional comments

Considering the authors' focus on Brazil, I am not sure if the title (tropical deforestation) is appropriate. Consider change it.

Indeed. We have suggested a title change which aligns with editor recommendations

Replace the ref. #35 by Azevedo-Ramos, C., Moutinho, P., Arruda, V. L. da S., Stabile, M. C. C., Alencar, A., Castro, I; Ribeiro, J. P. (2020). Lawless land in no man's land: The undesignated public forests in the Brazilian Amazon. *Land Use Policy* (January), 104863. <https://doi.org/10.1016/j.landusepol.2020.104863>.

Done.

Reviewers' Comments:

Reviewer #1:

Remarks to the Author:

The authors have sufficiently addressed my concerns and I support publication, though I'd still advocate as much explanation as possible of weighted results on the basis of raw matched results with highlighting when they appear to differ.

Reviewer #2:

Remarks to the Author:

I'm afraid I have to disagree with the authors' opinion on my suggestion to provide recommendations. Discussing potential implications of results published in scientific articles is more than necessary these days. Inferences - or recommendations - are welcome if the authors provide sufficient clarification on the limitations of the analysis performed. However, I accept and understand the author's decision not to include recommendations and consider the article to be suitable for publication by Nature Communication.

Responses to the Reviewers

Reviewer #1 (Remarks to the Author):

The authors have sufficiently addressed my concerns and I support publication, though I'd still advocate as much explanation as possible of weighted results on the basis of raw matched results with highlighting when they appear to differ.

We thank Reviewer 1 for their thoughtful comments and feedback. We are now publishing the unweighted and weighted estimates as a part of the Source Data file of the manuscript.

Reviewer #2 (Remarks to the Author):

I'm afraid I have to disagree with the authors' opinion on my suggestion to provide recommendations. Discussing potential implications of results published in scientific articles is more than necessary these days. Inferences - or recommendations - are welcome if the authors provide sufficient clarification on the limitations of the analysis performed. However, I accept and understand the author's decision not to include recommendations and consider the article to be suitable for publication by Nature Communication.

We completely agree with Reviewer 2 that the policy implications of research are of utmost importance. Nonetheless, we feel that we have provided policy recommendations as precisely as possible without delving into speculative interpretation of our results. Instead, we have proposed that the Editor commission a comment/perspective piece to be written by a Brazilian researcher to accompany our article. We believe this would be the appropriate avenue for such contextualized policy recommendations, particularly given the current political environment in Brazil.